# Benefit versus risk of chromosomal microarray analysis performed in pregnancies with normal and positive prenatal screening results: A retrospective study

Rami Moshonov[1][ID]*, Keren Hod[ID][2][ID], Bella Azaria[3], Ifat Abadi-Korek[2], Rachel Berger[4], Mordechai Shohat[4,5]

1 Obstetrics and Gynecology Department, Assuta Medical Centers Affiliated with Ben-Gurion University of the Negev, Tel Aviv, Israel, 2 Department of Academy and Research, Assuta Medical Centers Affiliated with Ben-Gurion University of the Negev, Tel Aviv, Israel, 3 Division of Medicine, Assuta Medical Centers Affiliated with Ben-Gurion University of the Negev, Tel Aviv, Israel, 4 The Genetic Institute of Maccabi Health Medicinal Organization, Rehovot, Israel, 5 Sackler School of Medicine, Tel Aviv University, Tel Aviv, Israel

☯ These authors contributed equally to this work.
* ramim@assuta.co.il

**Data Availability Statement:** Study data are only available upon request since the dataset includes sensitive medical information (i.e. genetic

## Abstract

### Background

Most studies on chromosomal microarray analysis (CMA) and amniocentesis risks have not evaluated pregnancies with low risk for genetic diseases; therefore, the efficacy and safety of CMA and amniocentesis in this population are unclear. This study aimed to examine the benefits and risks of prenatal genetic diagnostic tests in pregnancies having low risk for chromosomal diseases.

### Methods and findings

In this retrospective study, we used clinical data from a large database of 30,830 singleton pregnancies at gestational age 16–23 weeks who underwent amniocentesis for karyotyping with or without CMA. We collected socio-demographic, medical and obstetric information, along with prenatal screening, CMA and karyotyping results. Fetal loss events were also analysed. CMA was performed in 5,837 pregnancies with normal karyotype (CMA cohort). In this cohort, 4,174 women had normal prenatal screening results and the risk for identifying genetic abnormalities with >10% risk for intellectual disability by CMA was 1:102, with no significant difference between maternal age groups. The overall post-amniocentesis fetal loss rate was 1:1,401 for the entire cohort (n = 30,830) and 1:1,945 for the CMA cohort (n = 5,837). The main limitation of this study is the relatively short follow-up of 3 weeks, which may not have been sufficient for detecting all fetal loss events.

### Conclusion

The low risk for post-amniocentesis fetal loss, compared to the rate of severe genetic abnormalities detected by CMA, suggests that even pregnant women with normal prenatal screening results should consider amniocentesis with CMA.

information of embryos, post-amniocentesis fetal loss events, and pregnant women's decision to abort following karyotype/CMA findings). Furthermore, Israeli regulations on secondary use of digital health data (2018) restrict the sharing of de-identified datasets, and requires such information to be shared only within a framework of an agreement. Therefore, if inquiries are received, the data will be shared only after the signing of an agreement in accordance with these regulation. Data requests may be sent to the Clinical Research Unit of Assuta Medical Center (20 HaBarzel Street, Tel Aviv, Israel; Phone: 972-3-764-5491, RESEARCH@ASSUTA.CO.IL.

**Funding:** This work was supported in part by the Adler Chair of the Sackler School of Medicine, Tel-Aviv University (https://en-med.tau.ac.il/; grant number is not applicable). This general academic fund was awarded as an honorarium to Prof. Shohat (M.S.) and was used for administrative purposes in the current study (grant number is not applicable). The funders had no role in study design, data collection and analysis, decision to publish, or preparation of the manuscript.

**Competing interests:** The authors have declared that no competing interests exist.

## Introduction

Most pregnancies (95%) have normal prenatal screening test results (i.e. nuchal-translucency, first-and second-trimester biochemical screening, second trimester ultrasound organ scan, and/or an integrated test) [1]; hence they are considered at low risk for chromosomal diseases. In many cases, even women carrying low-risk pregnancies often debate whether they should undergo diagnostic procedures to confirm that their fetus does not have genetic abnormalities.

Prenatal genetic screening (i.e. nuchal-translucency, first-and second-trimester biochemical screening, second trimester ultrasound organ scan, and/or an integrated test) identify risks for genetic disorders in fetuses [2]. Karyotyping by amniocentesis is an invasive diagnostic test to identify prenatal diagnosis of trisomy 21 (Down syndrome) and other fetal chromosomal abnormalities [3]. Chromosomal microarray analysis (CMA) can identify most chromosomal abnormalities detected by karyotyping, as well as additional smaller unbalanced changes [4, 5].

Amniocentesis has been associated with an increased risk of fetal loss [6]. Although several studies have re-evaluated this risk, there was a great degree of heterogeneity among them [7, 8]. Therefore, reliable information on the risks and benefits of CMA is essential in order to allow women to make informed decisions about this procedure.

The Israeli national 'healthcare basket' covers most of the prenatal screening tests at minimum cost. Since 1993, the Israeli Ministry of Health has subsidized amniocentesis in all women aged ≥35 years as well as in women aged <35 years at high risk for chromosomal abnormalities (≥1:380) [9]. As a result, according to the last available report by the Ministry of Health, 47% of Israeli Jewish women aged ≥35 years and 11% of women aged <35 years underwent amniocentesis, compared to 5% in the Western world [10].

In this study we aimed 1) to evaluate the rates of positive CMA results in pregnancies with normal karyotype and to compare them by maternal age and by prenatal screening results (normal versus positive); and 2) to examine the post-amniocentesis fetal loss risk at mid-trimester.

## Materials and methods

### Setting and study population

This retrospective study was conducted at the amniocentesis unit at Assuta Medical Centers (Tel Aviv, Israel) after receiving approval from the institutional ethics committee (Helsinki Committee; study number: 100-16-ASMC).

Data on women carrying singleton pregnancies who underwent karyotyping by amniocentesis at the amniocentesis unit—either with or without CMA—at gestational age 16–23 weeks between June 1, 2010 and August 31, 2015 were included in this study.

The indications for amniocentesis are presented in S1 Table. The procedures were ultrasound guided (Pro-US Philips HD7), using 20-21G spinal needles without local anaesthetic to aspirate 20–40 ml of amniotic-fluid. All physicians performing the procedures were OB/GYN board certified, and they all used the same ultrasound guidance technique, the same equipment and the same medical staff. Samples were analysed in the same laboratory throughout the study period (Mega-Lab, Rehovot, Israel) according to the Illumina protocol [11]. All couples with abnormal findings in amniotic fluid received detailed genetic counselling by the same physician (board certified in paediatrics and medical genetics).

### Data collection

Data from the patients' medical files were entered into the database by professional typists who were trained by the investigators. Data entries were monitored for correctness by the co-

author (K.H.) by comparing most of them (>50%) to the medical records. All patient data were de-identified prior to the analysis.

Collected data included socio-demographic information, medical and obstetric information, gestational age, maternal age, a detailed second trimester ultrasound organ scan, and at least one of the following: nuchal translucency, first-trimester biochemical screening, second trimester screening, and/or an integrated test. In addition, complete genetic results of CMA, as well as fetal loss events and elective terminations of pregnancies following a positive CMA result were documented.

Pregnancies with normal prenatal screening results were defined as those having detailed normal second trimester ultrasound organ scan results without soft-markers (i.e. thickened nuchal fold, hyperechoic bowel, shortened limbs, echogenic intracardiac focus, choroid plexus cysts, pyelectasis and single umbilical artery) [2], nuchal-translucency ≤ 2.9 mm and Down syndrome risk below 1:380 (the standard threshold in Israel) [9] according to either first or second trimester tests, or according to integrated screening tests [12–14].

Pregnancies not meeting the criteria for normal prenatal screening were defined as "pregnancies with positive prenatal screening results". In addition, pregnancies meeting the following criteria were defined as having positive screening results: treatment with any drug that might cause chromosomal changes (e.g., colchicine, podophyllotoxin, 6-mercaptopurine, 5-fluorouracil, azathioprine or propylthiouracil); abnormal CMA or karyotype results in previous pregnancies; at least one of the parents was a carrier of a translocation or mosaic; high human-chorionic gonadotropin levels (>3 MOM), low pregnancy-associated plasma protein-A levels (<0.15 MOM), low estriol levels (<0.15 MOM); or a diagnosis of high risk for any genetic diseases, including Mendelian disorders.

## Study endpoints

**Analysis of the rates of genetic abnormalities detected by CMA in pregnancies with normal karyotypes, and normal or positive prenatal screening results.** This analysis was aimed at determining the rates of genetic abnormalities that can be detected by CMA in pregnancies with normal prenatal screening results, and to compare them to the rates of genetic abnormalities detected by CMA in pregnancies with positive prenatal screening results. In addition, the rates of genetic abnormalities were analysed by maternal age (<35 versus ≥ 35 years).

The analysis included a subset of the study population (termed "CMA cohort") that had a pregnancy with a normal karyotype (i.e., after ruling out chromosomal abnormalities) and decided to undergo this test after receiving an explanation about it.

A CMA result was considered abnormal if it was categorized as a pathogenic, or a likely-pathogenic known chromosomal number variation (CNV) with a risk of >10% for intellectual disability, or as a de-novo, previously undescribed, microdeletion/microduplication >1Mb, that contained at least three Online Mendelian Inheritance in Man (OMIM®)-morbid genes. Abnormal CMA results were further divided into 2 intellectual disability risk categories: >50% risk (severe) and 10%-50% risk (moderate). CMA results were not considered abnormal if they were identified as a known CNV with <10% penetrance, CNV of unknown significance (VUS), or likely-benign or benign CNV (Table 1) [15].

In addition, we analysed the rate of pregnancies that were electively terminated following a positive CMA result.

**Analysis of post-amniocentesis fetal loss risk at mid-trimester.** Next, we assessed the post-amniocentesis fetal loss rate in the entire database (n = 30,830) as well as in the CMA cohort (n = 5,837) as we also wanted to evaluate this rate in pregnancies with normal karyotype

**Table 1. CMA results not considered abnormal in the current study.**

| CNV with <10% penetrance, such as the common microdeletion (del)/microduplications (dup) in: |
|---|
| • 15q11.2 (NIPA1)–del/dup |
| • 15q13.3 (CHRNA7)—dup |
| • 16p13.11 (MYH11)–dup |
| • distal 16p11.2 (SH2B1)—dup |
| • 16p12.1 (CDR2) -dup |
| • 1q21.1 (RBM8A)–del/dup |
| • Charcot-Marie-Tooth disease type 1A (CMT1A)–del/dup |
| • Steroid sulfatase (STS) deficiency–del/dup |
| CNV of unknown significance (VUS), and likely-benign or benign CNV, including: |
| • Inherited previously undescribed microdeletions or microduplications of any size |
| • De-novo previously undescribed microdeletion/microduplication ≤1Mb or more than 1MB with less than 3 Online Mendelian Inheritance in Man (OMIM®)-morbid genes |

that underwent CMA. Post-amniocentesis fetal loss was defined as fetal loss or intrauterine demise within 3 weeks of the procedure with no other cause [6]. The work protocol at the amniocentesis unit mandates contacting each woman three weeks after the amniocentesis procedure in order to provide the test result, monitor her condition and to document any complications that might have occurred since the procedure.

## Statistical analysis

All data are presented as the mean ± standard deviation (SD) for continuous variables with normal distribution, as median (IQR) for continuous non-normally distributed variables, and as the number of patients (percentage) for categorical variables. The one-sample Kolmogorov–Smirnov test was used to verify which of the continuous variables were distributed normally. Categorical variables were compared using chi-squared and Fisher's exact tests. Continuous variables were compared using t-tests and Mann-Whitney tests.

Statistical analysis was performed using the SPSS statistical package (Version 26, IBM Inc., Armonk, N.Y.). All statistical tests were two-tailed and p values below 5% were considered statistically significant.

# Results

## Study population

Of a total of 30,830 amniocentesis procedures performed on singleton pregnancies in the amniocentesis unit during the study period (2010–2015), 6,218 pregnancies also had CMA. A total of 259 pregnancies (4.2%) were excluded from the analysis due to insufficient data on prenatal screening results (i.e. missing required data for the classification by prenatal screening test results: ultrasound organ screening, nuchal translucency and the first/second/integrated biochemical screening test).

The pregnancies that were eligible for this analysis (n = 5,959) were divided into three groups: 1) pregnancies with normal karyotypes, as termed "CMA cohort" (n = 5,837, 97.9%); 2) pregnancies with abnormal karyotypes (n = 101, 1.7%); and, 3) pregnancies with unknown karyotypes (n = 21, 0.4%). Each group was further divided into subgroups by prenatal screening results (normal versus positive) and maternal age group (<35 years versus ≥35 years), (Fig 1). The characteristics of the women carrying the pregnancies with normal karyotype and normal prenatal screening results are summarized in S2 Table.

Of note is that 48.21% of amniocentesis procedures (14,864/30,830) were performed although there was no medical indication for them other than maternal choice (S1 Table).

## Rate of genetic abnormalities detected by CMA in patients with a normal karyotype

To obtain a pure low-risk population, this analysis was done only on eligible pregnancies with a normal karyotype (n = 5,837); pregnancies that had an abnormal karyotype (n = 101) and an unknown karyotype (n = 21) were excluded. No statistically significant differences in the prevalence rate of positive abnormalities discovered by CMA in pregnancies with a normal karyotype were observed between pregnancies with normal prenatal screening results (1:102, n = 4,174) and pregnancies with positive prenatal screening results (1:79, n = 1,663). Moreover, maternal age did not affect the prevalence of genetic abnormalities detected by CMA in women with normal or positive prenatal screening results.

## The rate of pregnancies that were electively terminated following genetic abnormalities detected by CMA in patients with a normal karyotype

As shown in Table 2, a total of 42/5,837 (0.7%) pregnancies were electively terminated following a positive CMA result (24 pregnancies with normal prenatal screening results, and 18 pregnancies with positive prenatal results). Elective termination rates did not differ significantly between maternal age groups and between pregnancies with normal and positive prenatal screening results.

## Analysis of post-amniocentesis fetal loss rate

Most amniocentesis procedures (75.5%, 23,276/30,830) were performed by five physicians; each performed a median of 3,954 procedures during the study period (range, 2,148–10,673).

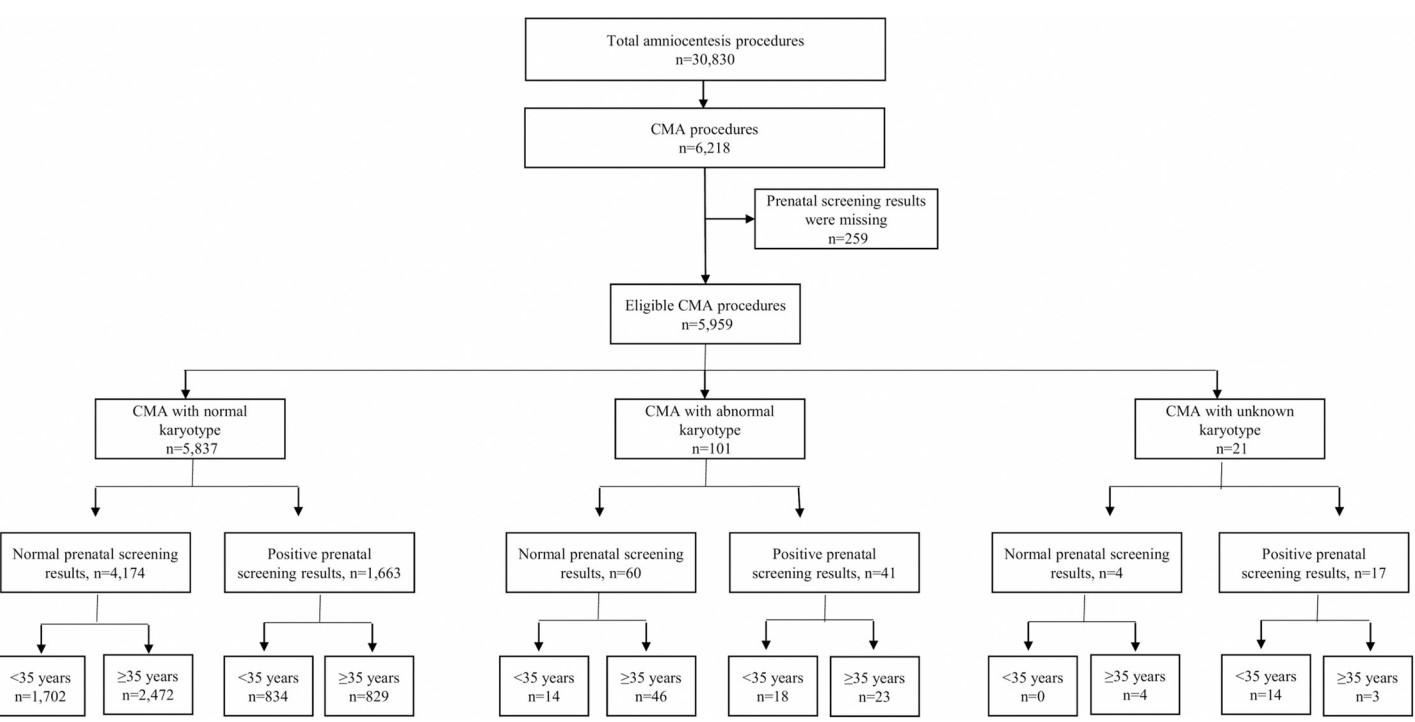

**Fig 1. Study flow chart.** Fig 1 showing the number of pregnancies analysed in each subgroup.

**Table 2. Abnormal CMA results and elective termination of pregnancies by maternal age in pregnancies with normal or positive prenatal screening results and normal karyotype (n = 5,837).**

| Outcome | Pregnancies with normal prenatal screening results | | | | Pregnancies with positive prenatal screening results | | | | |
|---|---|---|---|---|---|---|---|---|---|
| | <35 years | ≥35 years | Total | P-value[a] | <35 years | ≥35 years | Total | P-value[b] | P-value[c] |
| | n = 1,702 | n = 2,472 | n = 4,174 | | n = 834 | n = 829 | n = 1,663 | | |
| Total pathogenic abnormalities, n, (abnormal CMA rate) | 19 (1:89) | 22 (1:112) | 41 (1:102) | 0.533 | 7 (1:119) | 14 (1:59) | 21 (1:79) | 0.283 | 0.401 |
| %TOP (TOP/number of abnormal CMA) | 73.6% (14/19) | 45.4% (10/22) | 58.5% (24/41) | 0.229 | 71.4% (5/7) | 92.8% (13/14) | 85.7% (18/21) | 0.477 | 0.123 |

Abbreviations: CMA, chromosomal microarray analysis; TOP, termination of pregnancy.

[a]P-values for the comparison between maternal age groups of pregnancies with normal prenatal screening results.

[b]P-values for the comparison between maternal age groups of pregnancies with positive prenatal screening results.

[c]P-values for the comparison between pregnancies with normal versus positive prenatal screening results.

The rest of the physicians (n = 112) performed a median of 201 procedures (range, 1–665 procedures), which accounted for 24.5% (7,554/30,830) of all procedures.

Post-amniocentesis fetal loss rate in the entire database population was 1:1,401 (22/30,830) and 1:1,945 (3/5,837) in the CMA cohort. Fetal loss occurred at a median gestational age of 20 weeks (range, 17–22 weeks) in the entire database population and 18 weeks (range 17–19 weeks) in the CMA cohort, respectively. The median time between amniocentesis and fetal loss was 14 days (range, 1–21 days) and 8 days (range 7–9 days), respectively.

## Discussion

The current analysis has shown that the risk for severe genetic abnormalities detected by CMA in low-risk pregnancies, i.e., pregnancies with normal prenatal screening results and normal karyotype, is 1:102, regardless of maternal age. Furthermore, the rate of positive abnormalities discovered by CMA was similar in pregnancies with normal and positive prenatal screening results. The analysis also revealed a post-amniocentesis fetal loss rate of 1:1,401 in the entire cohort and 1:1,945 in the CMA cohort.

Almost half of amniocentesis procedures were performed without a medical indication. This finding is not surprising and is consistent with the growing trend of elective prenatal testing in Israel since the mid-1990s [16]. There are two main reasons for this trend. First, as mentioned above, the Israeli national 'healthcare basket' covers amniocentesis for women aged ≥35 years and women aged <35 years at high risk. Second, several key social influences drive pregnant women's choice of amniocentesis, as previously reported by Remennick [16].

CMA provides additional information over karyotyping in 6–7% of pregnancies in which abnormalities were identified by ultrasound [17–21]. It can identify relatively high rates of microdeletions and microduplications with a frequency of 1:10 in foetuses with malformations detected by ultrasound [22, 23]. Accordingly, the American Congress of Obstetricians and Gynecologists recommends CMA as the first tier test in the diagnostic evaluation of fetal structural abnormalities [24]. Conversely, there is limited information on abnormalities detection rate by CMA in pregnancies that have low risk for chromosomal diseases [25]. A recent meta-analysis showed that the frequency of CNVs associated with early onset syndromic disorders is 1:270 [25]. Approximately 1:909 cases involved late-onset diseases and 1:333 cases involved susceptibility CNV [25]. By adding the individual risk for pathogenic CNVs to the individual risk for cytogenetically visible chromosome aberrations, the overall risk for a clinically significant cytogenetic aberration was >1:180 [21]. Furthermore, pregnant women younger than 36

years have a higher risk for pathogenic CNVs than for Down syndrome [25]. In this study, although there were no statistically significant differences in the rate of abnormalities between younger and older women, pregnancies with positive prenatal screening results showed a trend for higher prevalence of abnormalities in older women compared to younger ones.

The incremental benefit of CMA over karyotyping in foetuses without ultrasound-identifiable abnormalities shows considerable variability. A large systematic review showed clinically significant findings by CMA in 1% of cases [26]. while the rate reported by single studies was 0.4% [27] -2.0% [28]. These differences probably arise from the different array platforms utilized, their resolution, and the different reporting practices of each laboratory. In the present study we used a single nucleotide polymorphism-detailed CMA [11, 15]. Although local definitions of the pathogenicity of specific results vary among laboratories and have changed over time, new knowledge and greater sharing of results in public databases have led to increased numbers of genomic regions that are definitively associated with disease, and to a decreased incidence of VUS [29].

In the current analysis, we evaluated only CMA findings with severe implications (>10% risk for intellectual disability) in pregnancies with normal prenatal screening results and have not considered VUS or findings with relatively low penetrance. Importantly, the vast majority of abnormal CMA findings, mostly those defined as severe, would not have been discoverable by NIPT. Recent studies have concluded that the procedure-related risks of post-amniocentesis fetal loss are much lower than currently cited. For example, Akolekar et al. reported that the pooled procedure-related fetal loss risk is 0.11% (95% CI, 0.04–0.26%) [7]. Our findings indicate that post-amniocentesis fetal loss rate is slightly lower (0.07%). However, as the occurrence of fetal loss in our cohort was only followed for up to 3 weeks after amniocentesis, this short period may not have been sufficient for detecting all fetal loss events. Tabor et al. [6], in her classic randomized clinical trial found that the median time from amniocentesis to fetal loss was 21.5 days (the same as our follow-up period), but with a range of 5–67 days. Most procedure-related pregnancy losses were reported to occur within 14 days (0.6%), before gestational age of 24 weeks (0.9%) and irrespective of gestation age (1.9%) [30]. Hence, fetal loss rate may have been underestimated by the 3-week cutoff used in our analysis.

An additional potential limitation may have resulted from analysing pregnancies that had only one soft marker or abnormal CMA or karyotype results in previous pregnancies within the same "high risk" category as other pregnancies with positive prenatal screening results. These may have lowered the risk of finding abnormalities using CMA; however no significant differences in the rate of CMA abnormalities were found (0.8% versus 1.1%, P = 0.578 for pregnancies that had only one soft marker versus other pregnancies with positive prenatal screening results; and 0.0% versus 1.3%, P>0.999, for pregnancies that had abnormal CMA or karyotype results in previous pregnancies versus other pregnancies with positive prenatal screening results). Furthermore, as the retrospective design of this study, we could only collect retrospective data of the early ultrasound organ scan, which was performed prior to amniocentesis (at 14–17 weeks of gestation). Therefore, we might have missed number of congenital malformations that can be diagnosed only with the late ultrasound organ scan, which is performed after amniocentesis (at 20–23 weeks of gestation, but not before 19 weeks or after 25 weeks of gestation). However, a considerable number of congenital malformations can be diagnosed by the early ultrasound organ scan. Last, we did not investigate the possible effect of ethnicity, which may affect CMA abnormalities and may be a potential confounder. Further research is needed to address the question as to whether ethnicity could be associated with our findings.

The strengths of this study include its large cohort and the use of prenatal screening test results to rigorously define pregnancies with normal prenatal screening results. Furthermore,

all karyotype and CMA results were analysed in the same laboratory providing consistency of test results. In addition, CMA has excellent diagnostic performance, with negligible false negative and false positive results.

## Conclusions

Although professional guidelines do not recommended offering CMA or karyotyping to women carrying pregnancies with low-risk for chromosomal diseases [31]. our findings suggest that these tests can detect a relatively high rate of genetic abnormalities, corroborating other reports which showed that the minimum risk detected by CMA in any pregnancy is at least 1:150, and is ultimately greater than 1% [32]. Considering the low post-amniocentesis fetal loss rate observed in our cohort, there is significant importance to raising awareness of this procedure among women of all ages who are interested in preventing diagnosable severe genetic abnormalities. Further multicentre validations are needed to change the current concept.

## Supporting information

**S1 Table. Indications for amniocentesis.** Almost half (48.22%) of the indications for amniocentesis in the study cohort were non-medical: 30.07% of women underwent amniocentesis due to advanced maternal age while 18.14% had no indication for this procedure. The most common reason for amniocentesis for a medical indication was abnormal prenatal screening results (26.04%).
(DOCX)

**S2 Table. Characteristics of pregnancies with normal karyotype and normal prenatal screening results for which amniocenteses were performed for CMA.** There were no differences between maternal age groups, except for higher weight on the day of the procedure and a higher prevalence of pregnancies that started out with multiple embryos, and fertility treatments in women ≥35 years compared to women <35 years. Gestational age on the day of the procedure was also significantly higher in women <35 years compared to women ≥35 years; however, it was not clinically significant. Paternal age, the number of pregnancies, and the number of deliveries were higher among women ≥35 years compared to women <35 years, as they are directly related to maternal age.
(DOCX)

## Author Contributions

**Conceptualization:** Rami Moshonov, Bella Azaria, Ifat Abadi-Korek, Mordechai Shohat.

**Data curation:** Rami Moshonov, Keren Hod, Rachel Berger.

**Formal analysis:** Keren Hod.

**Investigation:** Keren Hod, Ifat Abadi-Korek, Rachel Berger, Mordechai Shohat.

**Methodology:** Keren Hod, Mordechai Shohat.

**Project administration:** Rami Moshonov, Keren Hod.

**Supervision:** Bella Azaria, Ifat Abadi-Korek, Mordechai Shohat.

**Writing – original draft:** Rami Moshonov, Keren Hod, Mordechai Shohat.

**Writing – review & editing:** Rami Moshonov, Keren Hod, Bella Azaria, Ifat Abadi-Korek, Rachel Berger, Mordechai Shohat.

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
