## [Decision Letter · Decision Letter 0]

19 Jan 2021

PONE-D-20-38279

Benefit versus risk of chromosomal microarray analysis performed in pregnancies with normal and positive prenatal screening results: a retrospective study

PLOS ONE

Dear Dr. Hod,

Thank you for submitting your manuscript to PLOS ONE. After careful consideration, we feel that it has merit but does not fully meet PLOS ONE’s publication criteria as it currently stands. Therefore, we invite you to submit a revised version of the manuscript that addresses the points raised during the review process.

We look forward to receiving your revised manuscript.

Kind regards,

Antonio Simone Laganà, M.D., Ph.D.

Academic Editor

PLOS ONE

Additional Editor Comments:

The reviewers have expressed positive comments regarding your article, raising only few concerns. Considering this point, I invite authors to perform the required minor revisions.

Journal Requirements:

2.) Please ensure you have thoroughly discussed any potential limitations of this study within the Discussion section, including the potential impact of confounding factors.

3.) Thank you for stating the following in the Acknowledgments Section of your manuscript:

'This work was supported in part by the Adler Chair of the Sackler School of Medicine, 330

Tel-Aviv University.'

'The authors received no specific funding for this work.'

4.) We note that you have indicated that data from this study are available upon request. PLOS only allows data to be available upon request if there are legal or ethical restrictions on sharing data publicly. For information on unacceptable data access restrictions, please see http://journals.plos.org/plosone/s/data-availability#loc-unacceptable-data-access-restrictions.

Reviewers' comments:

Reviewer's Responses to Questions

**Comments to the Author**

1. Is the manuscript technically sound, and do the data support the conclusions?

Reviewer #1: Partly

Reviewer #2: Yes

2. Has the statistical analysis been performed appropriately and rigorously? 

Reviewer #1: I Don't Know

Reviewer #2: Yes

3. Have the authors made all data underlying the findings in their manuscript fully available?

Reviewer #1: Yes

Reviewer #2: Yes

4. Is the manuscript presented in an intelligible fashion and written in standard English?

Reviewer #1: Yes

Reviewer #2: Yes

5. Review Comments to the Author

Reviewer #1: 1. Results (line 230) – The authors provide an analysis of the post-amniocentesis fetal loss rate that is based on the total number of procedures performed from 2010-2015 (total = 30,830). This is not helpful as we are only interested in the losses within the study cohort of 5,837. Stating the fetal loss rate is 1:1401 is misleading, if not read very closely. Please change this analysis to reflect the true fetal loss rate of your study cohort only.

2. Results (line 236) – As described above, the authors state the median time between procedure and fetal loss was 14 days. This reflects the total among 30,830 procedures. They later also say their follow-up time only 3 weeks and insufficient. Mixing up the data in this way is confusing. Please only relay the follow-up time for the final study cohort of 5.837. Additionally, explain why your follow-up was only 3 weeks since the study time frame ended in 2015.

3. Results – What were the false negative and false positive rates for CMA? Were the results compared to a follow-up CMA on the child when born?

4. Results – Were any pregnancies terminated based on a positive CMA result?

5. Results (line 148) – The theoretical calculation of the rate of genetic abnormalities seems convoluted and unnecessary.

Reviewer #2: The accuracy of NIPT varies by disorder. In other words, the accuracy for detecting trisomy 21 is quite different from detecting a microdeletion. How were such differences accounted for when analyzing whether the CMA findings could be detected by NIPT?

Many women decide whether or not to proceed with amniocentesis based on second trimester fetal anatomy surveys which are typically performed at 20-22 weeks’ gestation. Most amniocentesis procedures are performed beforehand, at approximately 16-18 weeks. Can the authors please expand on the sonograms that were performed prior the amniocentesis procedures? Did women also have an additional sonogram subsequent to the amniocentesis? Were any anomalies found that were not detected beforehand? This is important because the rate of fetal anomalies is approximately 3-4% in the general population.

The authors analyzed a very robust dataset and the findings are largely generalizable. I’m not sure the conclusion “These findings required further validation in future studies” is warranted.

6. PLOS authors have the option to publish the peer review history of their article (what does this mean?). If published, this will include your full peer review and any attached files.

Reviewer #1: No

Reviewer #2: No

---

## [Author Response · Author response to Decision Letter 0]

18 Feb 2021

February 18, 2021

Professor Emily Chenette

Editor-in-Chief

PLOS ONE

MS TITLE: Benefit versus risk of chromosomal microarray analysis performed in pregnancies with normal and positive prenatal screening results: a retrospective study

Dear Professor Chenette,

Thank you for the opportunity to revise and resubmit our paper. We also wish to thank the reviewers for their thorough review and constructive comments and suggestions relating to our manuscript. In response, we have revised portions of our paper. We believe that the quality of the paper has been improved as a result of this revision.

The following is our point-by-point response to the reviewers’ comments along with our revised article. The changes to the manuscript are highlighted in yellow

Sincerely, 

Dr. Rami Moshonov, 

Obstetrics and Gynecology, Assuta Medical Centers

20 HaBarzel Street, Tel Aviv, Israel. 

Phone: +972-3-7644528, E-mail: ramim@assuta.co.il

Journal Requirements 

Response: As you have requested we verified that our manuscript meets PLOS ONE's style requirements including those for file naming. 

2) Please ensure you have thoroughly discussed any potential limitations of this study within the Discussion section, including the potential impact of confounding factors.

Response: As you have suggested we have ensured that the discussion section of the manuscript includes any potential limitations of this study, including the potential impact of confounding factors: 

Page 16, lines 290-293:

"… Last, we did not investigate the possible effect of ethnicity, which may affect CMA abnormalities and may be a potential confounder. Further research is needed to address the question as to whether ethnicity could be associated with our findings."

3.) Thank you for stating the following in the Acknowledgments Section of your manuscript: 'This work was supported in part by the Adler Chair of the Sackler School of Medicine, Tel-Aviv University.'

Please remove any funding-related text from the manuscript and let us know how you would like to update your Funding Statement. Currently, your Funding Statement reads as follows: 'The authors received no specific funding for this work.'

Response: Thank you for turning our attention to this misunderstanding. The Adler Chair of the Sackler School of Medicine, Tel-Aviv University, is a general academic fund used for printing and office requirements. This fund was awarded as an honorarium to Prof. Mordechai Shohat by Tel Aviv University. In the current study it was used for administrative purposes. The funders had no role in study design, data collection and analysis, decision to publish, or preparation of the manuscript.

No additional external funding was received for this study. We have added this information to the cover letter and deleted it from the Acknowledgments section. In addition, we amended the funding statement. 

4) We note that you have indicated that data from this study are available upon request. PLOS only allows data to be available upon request if there are legal or ethical restrictions on sharing data publicly. For information on unacceptable data access restrictions, please see http://journals.plos.org/plosone/s/data-availability#loc-unacceptable-data-access-restrictions.

Response: Indeed, we have mentioned that data from this study are available upon request since the dataset includes sensitive (pseudonymized) medical information (i.e. genetic information of embryos, post-amniocentesis fetal loss events, and pregnant women's decision to abort following karyotype/CMA findings). 

The Israeli regulations, which are detailed below, restrict the sharing of de-identified datasets, and require such information to be shared only within a framework of an agreement. Therefore, if inquiries are received, the data will be shared after the signing of an agreement in accordance with these regulation. We have added these restrictions to the cover letter.

The General Director of the Israeli Ministry of Health published two circulars referring specifically to secondary use of digital health data, as listed below:

• The General Director of the Israeli Ministry of Health Circular, dated 17 January 2018, regarding secondary uses of health data. And

• The General Director of the Israeli Ministry of Health Circular, dated 17 January 2018, regarding collaborations based on secondary uses of health data.

The above-mentioned circulars on secondary uses of health data state that medical data shared for secondary use (mainly research purposes) will be de-identified, and further set detailed conditions for privacy, medical confidentiality and data security, which include inter alia, a requirement to enter a data use agreement directed specifically at privacy and ethical use concerns (even in cases where identifying data was removed from the dataset (pseudonymization), but is still theoretically identifiable or refers to a single person, i.e. not an aggregated or cumulative set). Furthermore, any secondary use of health data for research purposes (and the way of access – within the institution or by a way of transfer) must be pre-approved by the institutional ethics committee and secondary use of medical data committee.

Contact information for a data access committee and ethics committee, to which data requests may be sent: the Clinical Research Unit of Assuta Medical Center. 20 HaBarzel Street, Tel Aviv, Israel. Phone: 972-3-764-5491, Email: RESEARCH@ASSUTA.CO.IL

Reviewer #1: 

1. Results (line 230) – The authors provide an analysis of the post-amniocentesis fetal loss rate that is based on the total number of procedures performed from 2010-2015 (total = 30,830). This is not helpful as we are only interested in the losses within the study cohort of 5,837. Stating the fetal loss rate is 1:1401 is misleading, if not read very closely. Please change this analysis to reflect the true fetal loss rate of your study cohort only.

2. Results (line 236) – As described above, the authors state the median time between procedure and fetal loss was 14 days. This reflects the total among 30,830 procedures. They later also say their follow-up time only 3 weeks and insufficient. Mixing up the data in this way is confusing. Please only relay the follow-up time for the final study cohort of 5,837. Additionally, explain why your follow-up was only 3 weeks since the study time frame ended in 2015.

Response: Thank you for your important remarks. Since the reviewer has raises this point in two comments, we realized that our explanation regarding the cohorts analyzed was not clear. Two main cohorts were analyzed in the study: the first cohort (n=30,830) comprised the entire database population and chosen for the analyses of post-amniocentesis fetal loss rate. The second cohort (n=5,837) is a subset of the first one and was used for analyzing the rate of abnormalities in pregnancies that underwent CMA. Following the reviewer’s comment we have emphasized these two cohorts in the paper, and also calculated the post-amniocentesis fetal loss rate and the follow-up time of the second study cohort. 

The study was retrospective and as such we analyzed medical records in our database between the years specified (2010-2015). However, for each record analyzed we only had follow-up data for 21 days after amniocentesis because, as described in the Methods section of the manuscript, according to the work protocol at the amniocentesis unit each woman was contacted three weeks after the amniocentesis procedure in order to provide her with the test result, monitor her condition and to document any complications that might have occurred since the procedure. 

We have amended the text as follows:

Abstract, "Methods and finding" section, lines 33-34:

"The overall post-amniocentesis fetal loss rate was 1:1,401 for the entire cohort (n=30,830) and 1:1,945 for the CMA cohort (n=5,837)."

Page 6, line 124:

The analysis included a subset of the study population (termed "CMA cohort") that had a pregnancy with a normal karyotype (i.e., after ruling out chromosomal abnormalities) and decided to undergo this test after receiving an explanation about it.

Page 8 lines 141-143:

"Next, we assessed the post-amniocentesis fetal loss rate in the entire database (n=30,830) as well as in the CMA cohort (n=5,837) as we also wanted to evaluate this rate in pregnancies with normal karyotype that underwent CMA. 

Page 9, line 168:

"The pregnancies that were eligible for analysis (n=5,959) were divided into three groups: 1) pregnancies with normal karyotypes, as termed "CMA cohort" (n=5,837, 97.9%);"

Page 13, lines 212-217:

"Post-amniocentesis fetal loss rate in the entire database population was 1:1,401 (22/30,830) and 1:1,945 (3/5,837) in the CMA cohort. Fetal loss occurred at a median gestational age of 20 weeks (range, 17-22 weeks) in the entire database population and 18 weeks (range 17-19 weeks) in the CMA cohort, respectively. The median time between amniocentesis and fetal loss was 14 days (range, 1-21 days) and 8 days (range 7-9 days), respectively.”

3. Results – What were the false negative and false positive rates for CMA? Were the results compared to a follow-up CMA on the child when born?

Response: Thank you for your question. CMA has negligible false negative and false positive values (Wright D. genetic testing and molecular biomarkers, 2016). This was added to the strengths of the study as follows:

Page 16, lines 297-298:

"In addition, CMA has excellent diagnostic performance, with negligible false negative and false positive results."

In addition, any positive CMA findings are verified in our lab by comparing the parents' polymorphism to rule out possible replacement of samples or maternal infection.

Furthermore, all CMA tests were performed in the same laboratory throughout the study period (Mega-Lab, Rehovot, Israel) and according to the Illumina protocol. The pathological findings are defined in page 6, lines 127-135: 

"A CMA result was considered abnormal if it was categorized as a pathogenic, or a likely-pathogenic known chromosomal number variation (CNV) with a risk of >10% for intellectual disability, or as a de-novo, previously undescribed, microdeletion/microduplication >1Mb, that contained at least three Online Mendelian Inheritance in Man (OMIM®)-morbid genes… CMA results were not considered abnormal if they were identified as a known CNV with <10% penetrance, CNV of unknown significance (VUS), or likely-benign or benign CNV (Table 1)."

Thank you for the suggestion regarding comparing the CMA results to those of the child after birth. Our results were not compared to a follow-up CMA of the child after birth. Such comparisons may be the subject of future studies. 

4. Results – Were any pregnancies terminated based on a positive CMA result?

Response: Thank you very much for your important question. Inspired by your comment we have analyzed this and found that 42 of 5,837 pregnancies (0.7%) were electively terminated following an abnormal CMA result. In addition, we have further examined the rates of electively terminated in pregnancies with positive and normal prenatal screening results by maternal age (<35 years versus ≥35). This analysis is presented in the new Table 2.

In addition we amended the text accordingly:

Page 5, lines 97-98:

"In addition, complete genetic results of CMA, as well as fetal loss events and elective terminations of pregnancies following a positive CMA result were documented."

Page 7, lines 136-137:

"In addition, we analysed the rate of pregnancies that were electively terminated following a positive CMA result."

Page 11, lines 190-197:

"The rate of pregnancies that were electively terminated following genetic abnormalities detected by CMA in patients with a normal karyotype 

As shown in Table 2, a total of 42/5,837 (0.7%) pregnancies were electively terminated following a positive CMA result (24 pregnancies with normal prenatal screening results, and 18 pregnancies with positive prenatal results). Elective termination rates did not differ significantly between maternal age groups and between pregnancies with normal and positive prenatal screening results."

5. Results (line 148) – The theoretical calculation of the rate of genetic abnormalities seems convoluted and unnecessary.

Response: We appreciate your suggestion, and therefore decided to remove this analysis from the paper.

Reviewer #2: 

1. The accuracy of NIPT varies by disorder. In other words, the accuracy for detecting trisomy 21 is quite different from detecting a microdeletion. How were such differences accounted for when analyzing whether the CMA findings could be detected by NIPT?

Response: Thank you for your important comment. The fact that detection of some of the disorders with NIPT is less accurate than that of Down-Syndrome highlights the advantage of CMA over NIPT. However, following another reviewer’s suggestion we have decided to remove the theoretical calculation of the rate of genetic abnormalities from the paper.

2. Many women decide whether or not to proceed with amniocentesis based on second trimester fetal anatomy surveys which are typically performed at 20-22 weeks’ gestation. Most amniocentesis procedures are performed beforehand, at approximately 16-18 weeks. Can the authors please expand on the sonograms that were performed prior the amniocentesis procedures? Did women also have an additional sonogram subsequent to the amniocentesis? Were any anomalies found that were not detected beforehand? This is important because the rate of fetal anomalies is approximately 3-4% in the general population.

Response: Thank you for your important remark. In Israel, the prenatal screening test includes two sonogram tests: 1) the early sonogram, which is performed at 14-17 weeks of gestation (before amniocentesis); and 2) the late sonogram, which is performed at 20-23 weeks of gestation, but not before 19 weeks or after 25 weeks of gestation (after amniocentesis). 

The early sonogram scans the organs of the fetus in order to detect abnormalities and anatomic findings in the early stages of pregnancy. Although the development of some of the organs has not yet been completed at this stage (and therefore a late sonogram should also be done), a considerable number of congenital malformations can be diagnosed.

The late sonogram is also important for women who have done an early sonogram, because there are systems that develop only after week 15 of gestation and there are malformations that develop slowly and are manifested only beyond week 20 of gestation.

In this retrospective study we collected data of pregnant women who underwent amniocentesis; therefore, we could only collect retrospective data of the early sonogram test, which was performed prior to amniocentesis. 

We added this to the study's limitations, as follows:

Page 16, lines 283-290:

"Furthermore, as the retrospective design of this study, we could only collect retrospective data of the early ultrasound organ scan, which was performed prior to amniocentesis (at 14-17 weeks of gestation). Therefore, we might have missed number of congenital malformations that can be diagnosed only with the late ultrasound organ scan, which is performed after amniocentesis (at 20-23 weeks of gestation, but not before 19 weeks or after 25 weeks of gestation). However, a considerable number of congenital malformations can be diagnosed by the early ultrasound organ scan."

3. The authors analyzed a very robust dataset and the findings are largely generalizable. I’m not sure the conclusion “These findings required further validation in future studies” is warranted.

Response: Thank you for your remark. We removed this sentence from the conclusions, as you have suggested

Again, we thank the reviewers for their valuable comments. We appreciation the time and effort invested in their excellent reviews. 

Sincerely, 

Dr. Rami Moshonov, 

Obstetrics and Gynecology, Assuta Medical Centers

20 HaBarzel Street, Tel Aviv, Israel. 

Phone: +972-3-7644528, E-mail: ramim@assuta.co.il

---

## [Decision Letter · Decision Letter 1]

13 Apr 2021

Benefit versus risk of chromosomal microarray analysis performed in pregnancies with normal and positive prenatal screening results: a retrospective study

PONE-D-20-38279R1

Dear Dr. Hod,

We’re pleased to inform you that your manuscript has been judged scientifically suitable for publication and will be formally accepted for publication once it meets all outstanding technical requirements.

Kind regards,

Giuseppe Novelli

Academic Editor

PLOS ONE

Additional Editor Comments (optional):

Reviewers' comments:

Reviewer's Responses to Questions

**Comments to the Author**

1. If the authors have adequately addressed your comments raised in a previous round of review and you feel that this manuscript is now acceptable for publication, you may indicate that here to bypass the “Comments to the Author” section, enter your conflict of interest statement in the “Confidential to Editor” section, and submit your "Accept" recommendation.

Reviewer #2: All comments have been addressed

2. Is the manuscript technically sound, and do the data support the conclusions?

Reviewer #2: Yes

3. Has the statistical analysis been performed appropriately and rigorously? 

Reviewer #2: Yes

4. Have the authors made all data underlying the findings in their manuscript fully available?

Reviewer #2: Yes

5. Is the manuscript presented in an intelligible fashion and written in standard English?

Reviewer #2: Yes

6. Review Comments to the Author

Reviewer #2: The authors appear to have appropriately addressed the reviewers' comments. The manuscript is improved as a result.

7. PLOS authors have the option to publish the peer review history of their article (what does this mean?). If published, this will include your full peer review and any attached files.

Reviewer #2: No

---

## [Editor Report · Acceptance letter]

15 Apr 2021

PONE-D-20-38279R1 

Benefit versus risk of chromosomal microarray analysis performed in pregnancies with normal and positive prenatal screening results: a retrospective study 

Dear Dr. Hod:

I'm pleased to inform you that your manuscript has been deemed suitable for publication in PLOS ONE. Congratulations! Your manuscript is now with our production department. 

Kind regards, 

on behalf of

Prof. Giuseppe Novelli 

Academic Editor

PLOS ONE